# Rotavirus A Infection Prevalence and Spatio-Temporal Genotype Shift among Under-Five Children in Amhara National Regional State, Ethiopia: A Multi-Center Cross-Sectional Study

**DOI:** 10.3390/vaccines12080866

**Published:** 2024-08-01

**Authors:** Debasu Damtie, Aschalew Gelaw, Yitayih Wondimeneh, Yetemwork Aleka, Maryssa K. Kick, Zemene Tigabu, Ulrich Sack, Zelalem H. Mekuria, Anastasia N. Vlasova, Belay Tessema

**Affiliations:** 1Department of Medical Microbiology, School of Biomedical and Laboratory Sciences, College of Medicine and Health Sciences, University of Gondar, Gondar, Ethiopia; aschalew3@gmail.com (A.G.); yitayihlab@gmail.com (Y.W.); bt1488@yahoo.com (B.T.); 2Department of Immunology and Molecular Biology, School of Biomedical and Laboratory Sciences, College of Medicine and Health Sciences, University of Gondar, Gondar, Ethiopia; yetemnice@gmail.com; 3Ohio State University Global One Health Initiative LLC, Eastern Africa Regional Office, Bole Road, Noah Plaza, 2nd Floor, Addis Ababa, Ethiopia; 4Center for Food Animal Health, Department of Animal Sciences, College of Food Agricultural and Environmental Sciences, The Ohio State University, Wooster, OH 44691, USA; kick.28@osu.edu; 5Institute of Clinical Immunology, Faculty of Medicine, University of Leipzig, 04103 Leipzig, Germany; ulrich.sack@medizin.uni-leipzig.de; 6Department of Veterinary Preventive Medicine, College of Veterinary Medicine, The Ohio State University, Columbus, OH 43210, USA; mekuria.3@osu.edu; 7Department of Pediatrics and Child Health, School of Medicine, College of Medicine and Health Sciences, University of Gondar, Gondar, Ethiopia; zemene.tigabu@gmail.com; 8Global One Health initiative (GOHi), The Ohio State University, Columbus, OH 43210, USA

**Keywords:** rotavirus A, diarrhea, genotype, under-five children, vaccine, Ethiopia

## Abstract

**Background**: Globally, rotavirus (RV) A (RVA) is the most common cause of severe and sometimes fatal diarrhea in young children. It is also the major cause of acute gastroenteritis among children in Ethiopia. Currently, the WHO has prequalified four RVA vaccines for universal childhood immunization. Ethiopia introduced the monovalent Rotarix vaccine into its national immunization program in 2013. Since then, only a few studies on the burden and genotype distribution of RVA infection post-vaccine introduction have been conducted (mostly at sentinel surveillance sites). Therefore, this study aimed to assess RVA prevalence and genotype distribution among children under five years in Ethiopia (February 2021–December 2022). **Methods**: This multi-center hospital-based cross-sectional study involved 537 diarrheic children under-five years old. Rotavirus A detection was conducted using a one-step reverse-transcriptase polymerase chain reaction (RT-PCR). Genotyping was conducted by Sanger sequencing of the VP7 (complete) and VP4 (partial) genes. Descriptive analysis and Pearson’s chi-squared test were carried out using SPSS version 29. Phylogenetic analysis with 1000 bootstrap replicates was performed using MEGA version 11 software. Statistical significance was set at *p* < 0.05 for all analyses. **Results**: The prevalence of RVA infection among diarrheic children was 17.5%. The most prevalent G-types identified were G3 (37%), the previously uncommon G12 (28%), and G1 (20%). The predominant P-types were P[8] (51%), P[6] (29%), and P[4] (14%). The three major G/P combinations observed were G3P[8] (32.8%), G12P[6] (28.4%), and G1P[8] (19.4%). Phylogenetic analysis revealed clustering of Ethiopian strains with the globally reported strains. Many strains exhibited amino acid differences in the VP4 (VP8* domain) and VP7 proteins compared to vaccine strains, potentially affecting virus neutralization. **Conclusions**: Despite the high RVA vaccination rate, the prevalence of RVA infection remains significant among diarrheic children in Ethiopia. There is an observable shift in circulating RVA genotypes from G1 to G3, alongside the emergence of unusual G/P genotype combinations such as G9P[4]. Many of these circulating RVA strains have shown amino acid substitutions that may allow for neutralization escape. Therefore, further studies are warranted to comprehend the emergence of these unusual RVA strains and the diverse factors influencing the vaccine’s diminished effectiveness in developing countries.

## 1. Introduction

Rotavirus A (RVA) is the predominant cause of severe acute gastroenteritis among infants and young children globally. In 2016 alone, RVA infections were responsible for over 128,500 deaths in children under the age of 5 with a significant proportion occurring in sub-Saharan Africa [1]. This virus exerts a disproportionate burden on resource-limited regions, with more than 80% of all RVA-related fatalities concentrated in South Asia and sub-Saharan Africa [2]. Remarkably, nearly every child worldwide contracts RVA by the age of 5, regardless of their geographic or economic circumstances [3].

Rotaviruses have a genome consisting of 11 double-stranded RNA segments surrounded by a triple-layered icosahedral protein capsid that belongs to the *Sedoreoviridae* family [4]. The viral genome encodes six structural proteins (VP1–VP4, VP6, and VP7) and six non-structural proteins (NSP1–NSP6) [5]. Viral protein 2 (VP2) makes the first layer of the virus, with each vertex having VP1 and VP3 protein copies. Viral protein 6 (VP6) forms the second layer. The structural glycoprotein VP7 and the spike protein VP4 form the outermost protein layer. Based on antigenic relationships of their VP6, rotaviruses (RVs) are classified into nine groups (A–D and F–J) [4,6,7,8]. The combination of VP7 and VP4 genes, each of which encodes surface proteins capable of eliciting neutralizing antibodies independently, are used to classify RV species A (RVA) into different G (glycoprotein) and P (protease-sensitive) genotypes, respectively [9,10]. Currently, there are 36 G and 51 P genotypes of RVA described in both human and animals [10,11,12].

Although four groups of RVs (A, B, C, and H) are known to cause acute gastroenteritis in humans, RVA accounts for more than 90% human disease [13]. The predominant RVA genotypes are G1, G2, G3, G4, and G9 in conjunction with P[8], P[6], and P[4] worldwide [13]. Nonetheless, studies have shown wide geographical variation in G- and P-type prevalence across continents, global temporal changes in the frequency of dominant strains and emergence of unusual P and G types and their combinations were reported [13,14,15,16,17].

Two effective RVA vaccines, a single-strain attenuated human RVA vaccine (Rotarix, GlaxoSmithKline Biologicals, Rixensart, Belgium) and a multi-strain bovine-human reassortant vaccine (RotaTeq, Merck Sharp & Dohme LLC, Rahway, NJ, USA), have been recommended by the World Health Organization (WHO) for routine immunization of infants since 2009 [18]. Ethiopia has introduced RVA vaccine 4 years after WHO’s recommendation in 2013 to its Expanded Program on Immunization (EPI). The purpose of the RVA vaccine introduction in Ethiopia was primarily to prevent childhood deaths and hospitalizations related to RVA infection. Two doses of a single-strain attenuated RVA vaccine (Rotarix) is orally administered at 6 and 10 weeks of age [19]. A national level study reported 56% uptake of a complete RVA vaccine schedule among children aged 12–23 months [20]. A recent study also reported 52.3% and 68.86% RVA full immunization coverage at national level and Amhara National Regional State, respectively [21].

Some hospital-based studies in Ethiopia showed RVA as a major cause of nonbacterial acute gastroenteritis in infants and young children accounting for 18–28% of acute gastroenteritis cases [22,23,24,25,26]. However, evidence about the magnitude and genetic diversity of the circulating RVA strains in the study area is limited. Moreover, the impact of the RVA vaccine introduction on the magnitude of RVA infection among under-five children and the emergence of vaccine breakthrough genotypes is not well understood. Therefore, this study aimed to assess the magnitude and vaccine breakthrough genotypes of RVA infection among diarrheic children in Amhara National Regional State, Ethiopia.

## 2. Materials and Methods

### 2.1. Study Design and Settings

The study was conducted in Amhara National Regional State, Ethiopia. A multi-center hospital based cross-sectional study design was employed involving three hospitals (University of Gondar Comprehensive Specialized Hospital, Felege Hiwot Comprehensive Specialized Hospital, and Debre Markos Comprehensive Specialized Hospital) in Amhara National Regional State. A total of 537 under-five children with diarrhea visiting the outpatient and inpatient departments of the three hospitals from February 2021 to December 2022 were enrolled in this study. More than 50% of samples were collected from Gondar between February 2021 and December 2021, while another 50% of samples were collected from Bahir Dar and Debre Markos between May 2021 and December 2022.

### 2.2. Sociodemographic and Clinical Data Collection

Socio-demographic and clinical data of the study participants were collected by trained nurses using semi-structured questionnaires. Moreover, the immunization status, clinical presentation, and nutritional status of the children were determined by accessing immunization cards, conducting physical evaluation, and recording anthropometric measurements, respectively.

### 2.3. Nutritional Assessment and Clinical Severity Scorning of Diarrhea

Nutritional status of children was determined by using the World Health Organization (WHO) child growth standards (available at: https://www.who.int/tools/child-growth-standards/standards (accessed on 5 September 2023)). Malnutrition was defined as z-score < −SD as assessed by height for age, weight for age, and weight for height for stunting, underweight, and wasting, respectively [27]. Clinical severity of diarrhea was assessed using the Vesikari clinical severity scoring system that considers diarrhea episodes/24 h, diarrhea duration in days, vomiting episodes/24 h, vomiting duration in days, body temperature, dehydration status, and type of treatment received into account. The Vesikari clinical severity scoring system has a maximum of 20 points. Based on the Vesikari clinical severity score, clinical severity of diarrhea was categorized as mild (<7), moderate (7–10), and severe (≥11) [28,29].

### 2.4. Sample Collection, Transport, and Storage

Approximately 2 g of formed stool (or 2 mL for diarrheic stool) was collected from each under-five diarrheic child in a sterile stool cup. The collected stool samples were transferred into 2 mL cryovials and stored at −20 °C onsite. Samples were then transported in cold boxes to the Immunology and Molecular Biology Laboratory at the School of Biomedical and Laboratory Sciences, University of Gondar for further laboratory analyses. Subsequently, the samples were stored at −80 °C until tested.

### 2.5. RNA Extraction and RVA Detection

Viral RNA extraction was performed using QIAamp Mini spin viral RNA extraction kit (Qiagen, Hilden, Germany). One-step RT-PCR kit (Bio-Rad, Hercules, CA, USA) was employed to amplify non-structural protein 3 (NSP3) gene of RVA using NSP3F-5′-ACCATCTACACATGACCCTC-3′ and NSP3R-5′-GGTCACATAACGCCCC-3′ primers. Eighteen microliters of One-step RT-PCR master mix was prepared per sample from iTaq Universal SYBR green reaction mix (2x) (10 µL), nuclease free water (4.75 µL), iScriptRT enzyme (0.25 µL), NSP3F and NSP3R primers (10 µM) (1.5 µL) each. The master mix was added to the respective RNA sample, positive control, no template control and negative extraction control wells in a PCR plate. Two microliters of sample and respective controls were added to the respective wells to make up a total reaction volume of 20 µL. A reaction condition involving reverse transcription at 50 °C for 30 min followed initial denaturation at 95 °C for 10 min and 40 cycles of denaturation at 94 °C, annealing at 56 °C and extension at 72 °C for 30 s each was set for amplifying the target gene. Melting curve analysis was added to the program (denaturation at 95 °C, annealing 56 °C, and denaturation at 95 °C for 15 s each) to check for non-specific amplificon and primer-dimer formation.

### 2.6. RNA Shipment for Genotyping and Sequencing

The rotavirus-positive RNA samples were shipped to the Ohio State University, USA for genotyping and sequencing. The sample preparation for shipment is described briefly as follows. The RNA samples were first dried and stored in GENTegra RNA tubes (GenTegra LLC, Pleasanton, CA, USA) before shipment. According to the RNArchive protocol, a volume of 40 µL of RNA was added to the GENTegra tubes. The tubes containing the RNA were incubated for 5 min at room temperature (21–25 °C). The RNA was then mixed by pipetting up and down 10 times to solubilize and mix in the RNArchive matrix. The RNA was dried by letting the tubes open in a biosafety cabinet for 24 h. After shipment to the OSU, the dried RNA in the tubes were reconstituted with equal volume (40 µL) of molecular biology grade water to recover the RNA. Prior to the utilization of the kit, it was evaluated for its performance by comparing the initial concentration of the RNA in the sample and RNA concentration after four weeks of storage at room temperature using the kit. The kit has shown excellent performance in terms of maintaining the RNA concentration after four weeks of storage at room/ambient temperature. Upon arrival, the samples were tested for RVA using TaqMan qRT-PCR to confirm the integrity of the sample during shipping and almost all samples tested positive with satisfactory Ct values (16.4 ± 7.4 SD).

### 2.7. VP7 (G) and VP4 (P) Based Genotyping PCR

To determine the predominantly circulating RVA genotypes, one-step RT-PCR was conducted for all RVA-positive samples using genotyping primers targeting the outer capsid VP7 and VP4 genes. Beg 9-5′-GGC TTT AAA AGA GAG AAT TTC CGTCTGG-3′ and End 9-5′-GGT CAC ATC ATA CAA TTC TAA TCTAAG-3′ primer pairs targeting 1062 bp of the VP7 gene were used. VP4F-5′-ATGGCTTCGCTCATTTATAGACA-3′ and con2R-5′-ATT TCG GAC CAT TTA TAA CC-3′ primer pairs amplifying 877 bp fragment of the VP4 gene were used to amplify RVA VP4 genes. SuperScript IV One-Step RT-PCR kit (Thermo Fisher Scientific, Waltham, MA, USA) was used to amplify the target genes. For a large volume of PCR reactions, 37.5 µL of 2x Platinum™ SuperFi™ RT-PCR Master Mix, 4.5 µL of the respective forward and reverse primers each, 0.75 µL of SuperScript™ IV RT Mix, 20.25 µL of nuclease free water, and 6 µL of template RNA treated with 1.5 µL of DMSO were used to obtain 75 µL of the amplification products. The reaction condition involves 50 °C for 10 min for reverse transcription, 98 °C for 2 min of initial denaturation followed by 40 cycles of denaturation at 98 °C for 10 s, annealing at 50 °C for 10 s, and extension at 72 °C for 30 s. Final extension at 72 °C for 5 min followed by a hold at 4 °C was included in the PCR program. The resulting amplicons were gel-purified and subjected to Sanger sequencing following the methods indicated in previously published works [26,30,31,32].

### 2.8. Gel Electrophoresis and Sequencing

The amplicons from the one step RT-PCR reaction were run on 1.5% agarose gel (Figure 1). The PCR amplicons derived from VP7 (1062 bp) (Figure 1A) and VP4 gene (877 bp) (Figure 1B) were extracted from the gel using QIAquick gel extraction kit (Qiagen, Hilden, Germany). The amplicon + primer mixes were shipped to the Ohio State University James Comprehensive Cancer Center (CCC) Genomic Shared Resources (GSR) Laboratory for sequencing. Sequencing was performed using Sanger dideoxy method.

### 2.9. Genotyping and Phylogenetic Analysis

The VP7 and VP4 sequence data were verified by Basic Local Alignment Search Tool for Nucleotides (BLASTN) analysis on the National Center for Biotechnology Information (NCBI) database and a web-based RVA Genotyping Tool Version 0.1 (available at: https://www.rivm.nl/mpf/typingtool/rotavirusa/job/462661131/ (accessed on 30 May 2024)) to determine the G-types and P-types of the sequenced RVA VP7 and VP4 genes, respectively [33]. Multiple sequence alignments were conducted using the CLUSTAL Omega tool integrated into the Molecular Evolutionary Genetics Analysis (MEGA) version 11 software. Subsequently, phylogenetic trees were constructed using maximum likelihood method and General Time Reversible model validated by 1000 bootstrap replicates as previously reported [34]. Sequence data generated in this study were deposited at GenBank and assigned accession numbers PQ001370–PQ001437 and PQ001438–PQ001502 for VP7 and VP4 gene sequences, respectively.

### 2.10. VP7 and VP4 Protein Modeling

The three-dimensional VP7 and VP4 protein structures of the circulating as well as the vaccine strains were predicted by SWISS-MODEL (available at: https://swissmodel.expasy.org/interactive (accessed on 2 June 2024)) [35]. Subsequently, molecular graphics images were produced using the UCSF Chimera package version 1.5.3 (available at: http://www.cgl.ucsf.edu/chimera (accessed on 2 June 2024)) from the Resource for Biocomputing, Visualization, and Informatics at the University of California, San Francisco (supported by NIH P41 RR-01081) [36]. Comparative structural analysis was conducted between the circulating and vaccine strains to understand the ability of the former to escape vaccine induced neutralization immunity in the study area.

### 2.11. Statistical Analysis

The sociodemographic, clinical, and laboratory data were analyzed by SPSS version 29 statistical software. The frequency and cross-tabulations were performed to summarize descriptive data. Pearson’s chi-squared test was used to investigate the association between outcome and explanatory variables. Significance was set at *p* < 0.05 for all statistical tests.

## 3. Results

### 3.1. Sociodemographic and Clinical Characteristics

A total of 537 children with acute gastroenteritis were enrolled into the study. The mean age of the study participants was 26.4 + 15.2 months ranging from 2 to 59 months old. The majority of children involved in this study were (i) immunized against RVA 524/537 (97.6%), (ii) from urban settings 496/537 (92%), and (iii) from outpatient department 496/537 (92.4%) (Table 1).

### 3.2. Rotavirus A Prevalence

The prevalence of RVA infection among diarrheic children was 94/537 (17.5%, 95%CI = 14.3–21%). The prevalence was the highest in Bahir Dar city 41/140 (29.3%), followed by Gondar city 44/261 (16.9%) and Debre Markos 9/136 (6.6%) (Figure 2).

Younger age (<24 months old) children were more affected 63/279 (22.58%) compared to older (24–59 months old) children 31/258 (12%). Nearly all the study participants were immunized 7/537 (1.3%) and 517/537 (96.3%) for incomplete and complete series, respectively. Only 13/537 (2.4%) of the study participants have never received RVA vaccine. Participant location (*p* = 0.001), age of the child (*p* = 0.04), vomiting (*p* = 0.002), sunken eyes (*p* = 0.019), being on intravenous fluid therapy (*p* = 0.026), Vesikari clinical severity (*p* = 0.006), being underweight (*p* = 0.005), and wasting (*p* = 0.007) were found to be significantly associated with RVA infection (Table 2).

The overall diarrheic cases were higher during summer (June, July, and August), which is considered rainy season in Ethiopia. However, the prevalence of RVA-associated diarrhea was relatively higher during spring (February, March, and April), ranging from 29.6 to 46.2% (Figure 3).

### 3.3. Genotypic Distribution of RVAs

Based on the VP7 and VP4 gene sequences of the circulating RVAs, their G and P-types, respectively, were determined. Of the 94 positive samples, 71 (75.5%) for the VP7 and 67 (71.3%) for the VP4 were successfully genotyped. The circulating G-types identified were G1, G2, G3, G9, and G12 while the P-types included P[4], P[6], and P[8]. G3 was the most dominant G-type, detected in 26 (37%) followed by G12 in 20 (28%) and G1 in 14 (20%) of the samples (Figure 4A). The proportions of circulating P-types were P[8] 36 (51%), P[6] 21 (29%), and P[4] 10 (14%) (Figure 4B).

Eight different G/P combinations were identified. Of these, G3P[8] 22 (32.8%), G12P[6] 19 (28.4%), and G1P[8] 13 (19.4%) were relatively more frequent (Table 3). These three G/P combinations account for 80.6% of the circulating strains.

The diversity and distribution of RVA genotypes were different across the three sampling locations. G3, G12, and G9 were the dominantly circulating RVA genotypes in Gondar, Bahir Dar, and Debre Markos, respectively (*p* = 0.001). G2 and G9 were only detected in Gondar and Debre Markos, respectively. All the three P-types were detected in samples from Gondar and P[6] was predominantly detected in Bahir Dar. G3P[8], G12P[6], and G9P[4] are the predominantly circulating G/P combinations in Gondar, Bahir Dar, and Debre Markos, respectively (Figure 5).

### 3.4. Phylogenetic Analysis of VP7 Gene of the Circulating RVA Strains

Phylogenetic trees were constructed for 68 complete VP7 gene sequences of the circulating RVA strains. All the 13 G1 strains clustered together and were associated with P[8] except one, which was associated with the P[6]. There was high nucleotide identity among the circulating G1 strains (98.88–100%). Additionally, the circulating strains have shown high nucleotide identity with the Rotarix G1 strain (96.22–96.63%). These strains were also closely related to the wild-type human G1 strains reported from India, Pakistan, Russia, Iran, and the Rotarix G1 strain (Figure 6A).

All G2 strains were associated with P[4] and were identified in Gondar only. The circulating G2 strains were 100% identical to each other and clustered closely with the wild-type human G2P[4] strains from USA, Brazil, Kenya, Bangladesh, and a previous Ethiopian strain isolated in 2016 (Figure 6B).

Twenty one G3 genotypes were associated with P[8], while three G3 genotypes were associated with untypeable P-types. The nucleotide identity among the circulating G3 strains ranged from 91.23 to 100%. These G3 strains were grouped into two sub-clusters. The first sub-cluster included the strains identified in children from Gondar that were closely related to the previously reported G3P[8] Ethiopian strain, while the other half of the G3 strains predominantly from Bahir Dar clustered with the wild-type human RVA G3 strains from USA and Pakistan (Figure 6C).

There were six G9 strains in this study, and all of them were associated with P[4] except one that was associated with P[6]. The circulating G9 strains were shown to share high sequence identity among each other (99.69–100%). All the G9 strains were identified from Debre Markos. The G9 strains were shown to form a tight cluster with the wild-type G9P[4] strains of USA and Ghana. However, these strains seemed to be only distantly related to the previously reported wild-type G9P[8] strain from Bahir Dar, Ethiopia in 2016 (Figure 6D).

Twenty G12 RVA strains were identified in this study. The sequence identity among the circulating G12 strains ranged from 96.02 to 100%. The majority were from Bahir Dar (17/20) and were associated with P[6] except one associated with P[8]. These G12 strains were clustered into two distinct sub-clusters: the first sub-cluster comprised eighteen strains (seventeen from Bahir Dar and one from Debre Markos), and the second sub-cluster was a cluster of two strains both from Gondar. Unlike the first cluster, the second cluster was shown to be closely related to the human wild-type RVA strains previously reported from USA, Mozambique, and Mexico. However, none of the current G12 strains were closely related to G12 strains previously reported in 2010, 2012, and 2016 from Ethiopia (Figure 6E).

### 3.5. Phylogenetic Analysis of VP4 Gene of the Circulating RVA Strains

Phylogenetic analysis was performed for the VP4 sequences of the 65 RVA strains circulating in Amhara National Regional State, Ethiopia. The analysis grouped the P[4] strains into two closely related clusters. Interestingly, the P[4] sequences detected in association with G2 were clustered together and similarly, the sequences associated with G9 also clustered together, suggestive of sequence-specific inter-segment interactions between RVA genomic segments. The sequence identity between the circulating P[4] strains ranged from 96.79 to 100% among each other. The G9-associated P[4] strains are closely related to the wild-type G9P[4] human RVA strains reported from India and Kenya. On the other hand, G2-associated P[4] strains were shown to be closely related to the wild-type G2P[4] human RVA strains reported from South Korea and Belarus (Figure 7A).

Of the twenty-two P[6] genotypes, twenty were associated with G12 while the rest of the two P[6] genotypes were associated with G1 and G9, one each. The sequences of the circulating P[6] strains shared high nucleotide identity amongst each other, ranging from 98.15 to 100%. These P[6] strains formed a distinct cluster and were not related to previously reported strains in Ethiopia. A P[6] strain from Debre Markos clustered with the porcine-human reassortant G4P[6] and wild-type human G4P[6] strains isolated from children in China and Thailand, respectively (Figure 7B).

The majority of the P-genotypes were P[8] mostly associated with G3 and G1. The sequence identity among the circulating P[8] strains ranged from 88.23 to 100%, while with the Rotarix P[8] strain ranged from 88.59 to 91.26%. The P[8] strains were grouped into three clusters. The strains in the first cluster were associated with G1 and closely related to the Rotarix P[8] strain and wild-type human RVA G1P[8] strain reported from the USA. The remaining two clusters were associated with G3 genotype, but clustered based on the study location. One of the clusters of G3-associated P[8] strains was dominated by strains from Bahir Dar, while the other was dominated by strains from Gondar. The G3-associated P[8] strains were not closely related to the Rotarix vaccine P[8] strain. However, the dominant strains from Gondar were closely related to strains reported from Greece, Pakistan, and Russia, while the dominant G3-associated P[8] strains from Bahir Dar were closely related to strains reported from Kenya, Rwanda, Cameroon, USA, Zimbabwe, and Ethiopian P[8] strains associated with G3, G9, and G12 reported in 2010 and 2016 (Figure 7C).

### 3.6. Comparison of the VP7 Antigenic Epitopes with Vaccine Strains

Amino acid sequences spanning the three major antigenic epitopes (7-1a, 7-1b, and 7-2) of the VP7 protein were compared between the identified RVA strains and Rotarix and RotaTeq vaccine strains, revealing substantial amino acid sequence heterogeneity.

All circulating G1 strains (N = 13) showed 100% identity with the Rotarix G1 vaccine strain in the epitopes 7-1a, 7-1b, and 7-2. The circulating G1P[8] strains maintained intact VP7 protein antigenic epitopes similar to the Rotarix vaccine G1 strain resulting in their clustering as a G1-II lineage. For the current Ethiopian G2 strains (N = 5), there were 18 amino acid substitutions out of 29 residues compared to the Rotarix G1 vaccine strain and 13 substitutions compared to the RotaTeq G2 strain. A comparison of the circulating G3 strains revealed amino acid differences ranging from 12 to 14 out of 29 amino acids distributed across all the three antigenic epitopes compared to the Rotarix G1 strain. All circulating G9 strains (n = 6) showed 12/29 and 13/29 amino acid differences compared to RotaTeq G3 and Rotarix G1 vaccine strains, respectively, distributed throughout the three antigenic epitopes. However, compared to the RotaTeq G1 vaccine strain (G1-III), two substitution mutations (D97E and S147N) were observed in the epitopes 7-1a and 7-2 of the circulating G1 strains, respectively. Among the twenty-four circulating G3 strains, twelve showed 2/29 amino acid substitutions (K238N and D242N), eleven showed 3/29 (T91N, K238N, D242N), and one showed 4/29 (T91N, N94D, K238N, D242N) across 7-1a and 7-1b epitopes compared to RotaTeq G3.

Thus, the highest antigenic epitope variation was observed among circulating G12 strains compared to the G4 strain of RotaTeq (19/29) and G1 strain of Rotarix (17/29) consistent with the increased phylogenetic distance between the circulating G12 and the vaccine strains. Almost all strains with amino acid substitutions involved at least one substitution associated with antibody neutralization escape (Table 4 and Figure 8).

### 3.7. Comparison of VP4 Antigenic Epitopes with Vaccine Strains

Comparative amino acid sequence analyses of the four neutralizing antigenic epitopes (epitope 8-1, epitope 8-2, epitope 8-3, and epitope 8-4) of the VP8* component of VP4 in the circulating RVA strains and the two vaccine strains exhibited significant amino acid substitutions associated with antibody neutralization escape.

Of the eleven circulating lineage IV P[8] strains associated with G1, ten showed four amino acid substitutions (N113D, N192D, N194T, and N195S) out of twenty-eight residues at epitopes 8-1, 8-2, and 8-3, while one strain exhibited only two amino acid substitutions (N194T and N195S) at epitope 8-2 compared to Rotarix P[8] (P[8]-I). P[8] strains associated with G3 showed more amino acid substitutions to the VP8* region of the Rotarix P[8] strain than compared to the P[8] strains associated with G1. Thirteen out of the twenty-two lineage III P[8] strains associated with G3 had six amino acid substitutions (E150D, N194D, N195G, S125N, S131N, and N135D), while the remaining nine had five amino acid substitutions (E150D, N195G, S125N, S131N/R, and N135D) were distributed across the three antigenic epitopes except epitope 8-4 of the Rotarix P[8] strain. Conversely, more amino acid substitutions were observed in the circulating lineage IV P[8] strains associated with G1 than lineage III P[8] strains associated with G3 compared to the RotaTeq P[8] (P[8]-II) vaccine strain. The circulating P[8] strains associated with G1 have shown a total of 7/28 amino acid substitutions (N113D, N192D, N194T, D195S, N125S, R131S, and D135N) across the three neutralizing antigenic epitopes (8-1, 8-2, and 8-3). However, relatively fewer amino acid variabilities were observed between circulating P[8] strains associated with G3 and the RotaTeq P[8] strain. Thirteen strains showed variability in four out of twenty-eight amino acids (E150D, N194D, D195G, and R131N), nine strains exhibited variability in three out of twenty-eight amino acids (E150D, D195G, and R131N), and one strain displayed variability in two out of twenty-eight amino acids (E150D and D195G).

Comparative analysis of the amino acid sequences in circulating P[4] and P[6] strains showed 10/28 and 18/28 amino acid variabilities, respectively, compared to the Rotarix P[8] strain. These strains have shown the highest heterogeneity (25/28 and 24/28 amino acid variabilities) compared to the RotaTeq P[8] strain (Table 5 and Figure 9).

## 4. Discussion

In this study, the prevalence of RVA infection among diarrheic children was 93/537 (17.5%). This prevalence is lower than in previous studies from Ethiopia before [17,37,38] and after the introduction of RVA vaccine (Rotarix) [26]. Nonetheless, at least one post vaccine era sentinel surveillance study in Ethiopia have reported comparable findings [37]. Reduction in RVA prevalence post vaccine introduction has been reported elsewhere [39,40,41], signifying the role of the vaccine in the reduction in RVA-associated acute gastroenteritis among under-five children.

This study identified significant geographic variability in RVA prevalence and genetic diversity among children, with Bahir Dar showing the highest prevalence at 28.6% (with G12P[6] strains being dominant), followed by Gondar at 16.9% (with G3P[8] strains being dominant), and Debre Markos at 6.6% (with G9P[4] strains being dominant). This striking variability may result from differences in sampling timing, variable climatic conditions, behavioral, and other location-specific factors. Additionally, the prevalence of symptomatic RVA infections was higher among younger children (22.58%) compared to older children (12%), possibly due to their less developed immune systems [42] and limited previous exposure to RVA [43], aligning with findings from Nigeria and Brazil where RVA is more prevalent among children under 24 months old [44,45,46]. In this study, though, there was a relative increase in the prevalence of RVA in spring (February, March, and April). RVA has circulated year-round in the study area, emphasizing the endemicity of RVA in Ethiopia. This observation is supported by a study which reported a year-round disease pattern in low- and low-middle income countries [47]. However, studies also reported the seasonal nature of RVA infection in which the highest peaks of RVA prevalence happened during the dry and cooler seasons [48,49].

In this study, RVA infection correlated with severe clinical signs such as vomiting, sunken eyes, intravenous fluid treatment, and high Vesikari clinical severity scores. These findings are consistent with other studies linking the severity of diarrhea to RVA infection [50,51,52]. Additionally, RVA infection was associated with acute malnutrition, as evidenced by higher rates of wasting and underweight status. Similar associations between diarrhea and malnutrition have been reported in studies from Bangladesh [53] and Sudan [54].

Based on the VP7 gene sequence analysis of 71 RVA-positive samples, five distinct G-types (G1, G2, G3, G9, and G12) were identified, with G3 being the most prevalent (37%), followed by G12 (28%) and G1 (20%). This trend aligns with a previous meta-analyses study in Ethiopia indicating an increase in the prevalence of G3 strains post-Rotarix vaccine introduction, possibly due to the vaccine-driven selective pressure on G1 strains of the same lineage (G1-II) [55]. Similar observations were noted in Australia and Colombia, where an increase in G3P[8] strain prevalence was observed following Rotarix vaccine introduction [56,57]. Despite its global distribution, G4 genotype was not identified in this study, consistent with previous Ethiopian studies [17,26,37,38,55,58]. The circulating RVA strains also included P[4], P[6], P[8], and four untypable P-types, with P[8] being the most common (51%). This finding is consistent with several previous studies in Ethiopia [17,26,37,38,58] and elsewhere including Cameroon [59], Kenya [60], and China [61]. Major G/P combinations including G3P[8] (32.8%), G12P[6] (28.4%), and G1P[8] (19.4%) reflected a shift towards G3P[8], G12P[8], G9P[8], and other unusual combinations observed in our previous meta-analysis study [55]. Similar trends were reported in Mozambique [62] and China [63] post-Rotarix vaccine introduction, indicating the vaccine’s influence on RVA genotype distribution and emergence of new genotypes. Thus, the vaccine-induced immunity continues to impact the prevalence of dominant and emerging genotypes highlighting that ongoing surveillance and vaccine efficacy monitoring are crucial for RVA control strategies.

Our observations of the variable predominant G/P combinations circulating in Gondar (G3P[8]), Bahir Dar (G12P[6]), and Debre Markos (G9P[4]) were similar to those reported in proximate study locations previously [26].

Phylogenetic analysis of the currently circulating Ethiopian strains revealed that most strains are closely related to the previously reported global and local RVA variants. However, the dominant circulating RVA strains seem to be changing over time. G3P[8] and the unusual G12P[6] and G9P[4] tend to rise in the post-vaccine introduction era. In general, the sequences from Ethiopia during the post-vaccine period formed distinct clusters as well as clusters with other globally reported strains on the global phylogeny. This indicates that distinct virus populations as well as imported strains were circulating during this period. Some Ethiopian strains such as G12 P[6] clustered together in a separate branch, distinct from other global strains, suggesting that local strains observed were likely persisting within the study area or neighboring regions, accumulating genetic changes over time. Moreover, some circulating Ethiopian strains such as G1P[8] and G12P[6] are not related to the previously reported Ethiopian strains. This might be either due to importation of new strains from other countries or due to significant mutations through random point mutations and intra-genotype reassortment in previously circulating strains, resulting in distinct clustering patterns of the circulating strains [64].

In this study, a comparative protein sequence analysis of the antigenic epitopes of VP7 and VP4 proteins revealed significant amino acid residue variability between the circulating Ethiopian RVA strains and the monovalent Rotarix and the pentavalent RotaTeq vaccine strains. Similar observation of high antigenic epitope variability between the circulating RVA strains and the vaccine strains were reported from China [65], Qatar [66], Belgium [67], and Gabone [68]. This might be one of the reasons to have significant prevalence of RVA infection despite high RVA immunization rate in the study area.

Similar to a previous study from Zambia [69], the circulating G1P[8] strains in our study belonged to the same lineage (G1-II) as the Rotarix vaccine G1 strain, and their VP7 antigenic epitopes were indistinguishable. Expectedly, G2, G3, G9, and G12 strains showed a significantly higher amino acid variability in the VP7 antigenic epitopes compared to Rotarix, suggesting that the vaccine-induced immunity could be inadequate to control heterotypic (non-G1) strains [69,70]. Although the multivalent RotaTeq vaccine (that was not a part of the EPI in Ethiopia) could provide a broader [homotypic (against G1, G2, and G3 strains) and heterotypic (against G9 and G12 strains)] immunity, the circulating Ethiopian strains belonged to different lineages (G1-II, G2-IV, and G3-I) than those in RotaTeq vaccine (G1-III, G2-II, and G3-II). This was associated with multiple amino acid mismatches, including substitutions linked to neutralization escape, as seen in studies from the USA [71] and elsewhere [65,66,67,68]. Thus, it remains unclear if RotaTeq vaccine would provide a superior protection against the strains circulating in Ethiopia compared to that induced by the Rotarix vaccine.

The circulating strains belonged to different lineages (P[8]-III and P[8]-IV) than Rotarix (P[8]-I) or RotaTeq (P[8]-II), displaying numerous amino acid substitutions in the VP8* component of VP4 [containing four neutralizing antigenic epitopes (8-1, 8-2, 8-3, and 8-4)] of the circulating RVA strains, which can enable neutralization escape. However, lineage III P[8] strains exhibited fewer substitutions than lineage IV strains relative to Rotarix, while the opposite trend was observed for RotaTeq vaccine, consistent with findings from Gabon [68] and Qatar [66]. These substitutions in the VP8* region may compromise neutralizing immunity, posing a risk of immune evasion [72]. The continuous dominance of the P[8] strains observed in our and other studies despite the vaccine-associated homotypic immunity suggests its increased genetic plasticity compared to the common G types. Comparative analyses with heterotypic strains (P[4] and P[6]) also revealed substantial amino acid variations across VP8* epitopes, suggesting a high likelihood of neutralization escape from both homotypic and less effective heterotypic immunity against VP4 proteins by P[8] and non-P[8] strains, respectively [66,67,68].

## 5. Conclusions and Recommendation

Despite the high rate of RVA immunization in the study area, RVA infections among diarrheic children remain considerably prevalent. The predominantly circulating RVA genotypes in the study area appear to change over time post-vaccine introduction. Phylogenetic analysis revealed that different RVA strains cluster with strains reported elsewhere globally. However, strong spatial genotype variations were also confirmed in this study. The circulating RVA strains demonstrate significant variability in amino acid residues, including those involved in neutralization escape within the VP7 and VP4 antigenic epitopes, compared to the two widely RVA vaccine strains licensed for global use. This finding underscores the importance of continued surveillance, vaccine updating and development, and potentially adapting immunization strategies to better match the evolving diversity of RVA strains globally.

## Figures and Tables

**Figure 1 vaccines-12-00866-f001:**
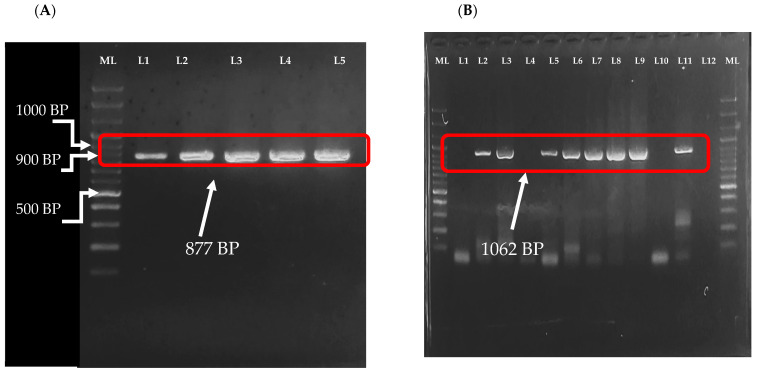
Gel image of RVA VP4 and VP7 gene amplicons. Viral protein 4 gene amplicon (**A**): ML is a molecular ladder of 100 plus base pair, L1 to L5 are representative samples with amplified VP4 gene with 877 bp size. Viral protein 7 gene amplicon (**B**). ML is a molecular ladder of 100 plus base pairs, L1, L4, L10, and L12 are lanes with no VP7 gene amplification while the rest of the lanes have the expected VP7 gene amplicon with 1062 base pair size.

**Figure 2 vaccines-12-00866-f002:**
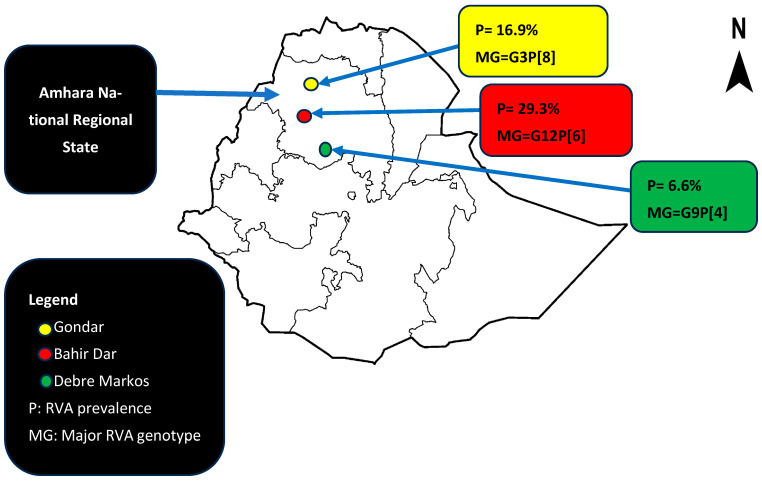
Spatial distribution of RVA infection among under-five children in Amhara National Regional State, Ethiopia.

**Figure 3 vaccines-12-00866-f003:**
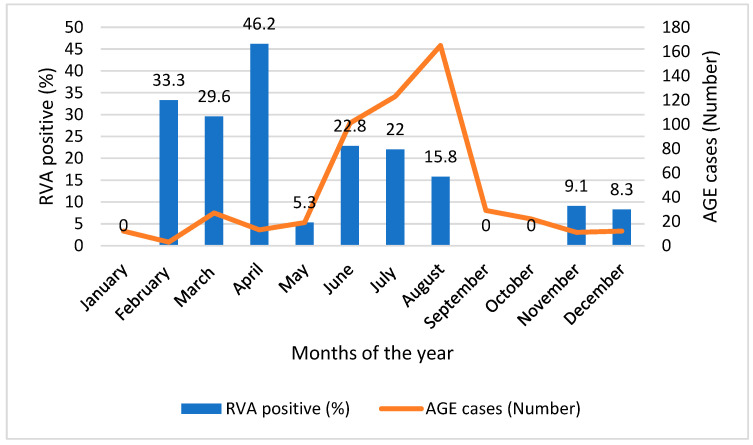
The distribution of RVA-positive cases by months of the year. The bars represent the prevalence of RVA infection by each month while the line graph represents the number of AGE cases enrolled in the study in each month.

**Figure 4 vaccines-12-00866-f004:**
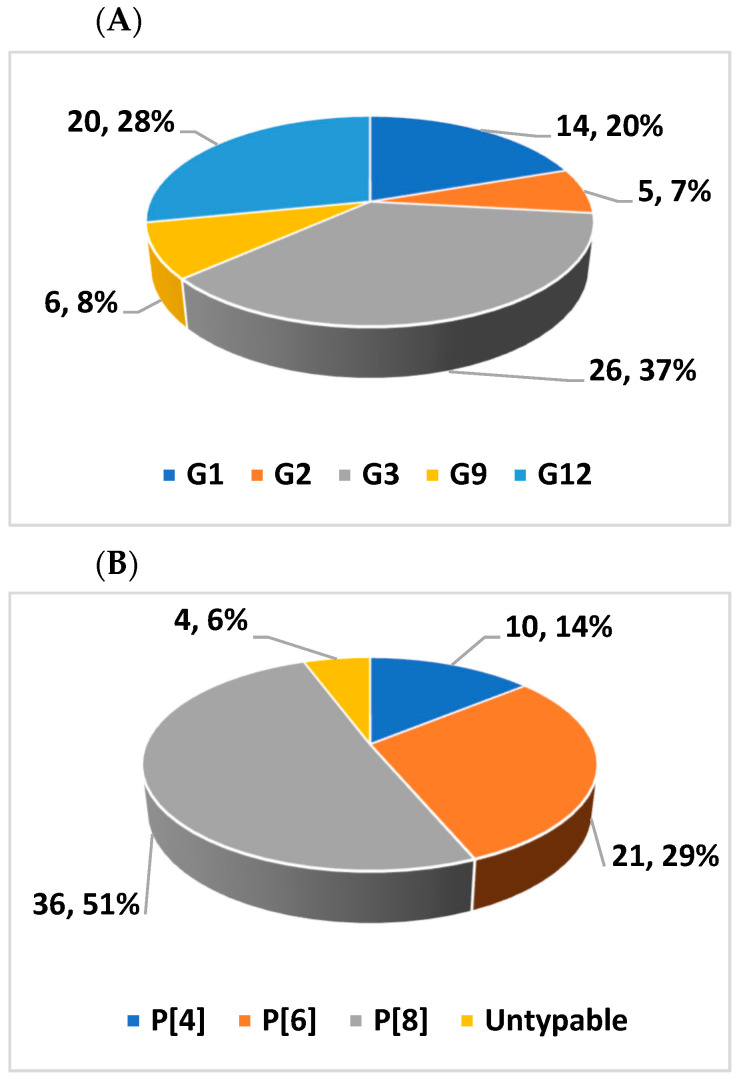
Distribution of RVA G types (**A**) and P types (**B**) isolated from children with acute gastroenteritis in Amhara National Regional State, Ethiopia February 2021–December 2022.

**Figure 5 vaccines-12-00866-f005:**
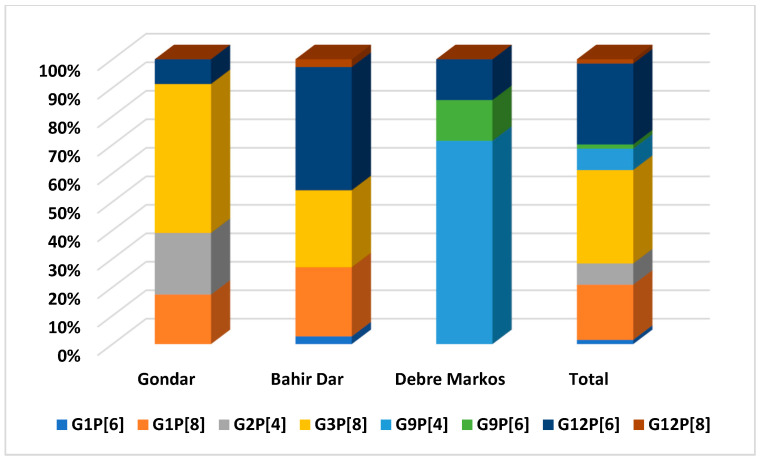
The circulating RVA G/P combination by study locations in Amhara National Regional State, Ethiopia, February 2021–December 2022. The *X*-axis represents the study locations, while the *Y*-axis represents the proportion of G/P combinations.

**Figure 6 vaccines-12-00866-f006:**
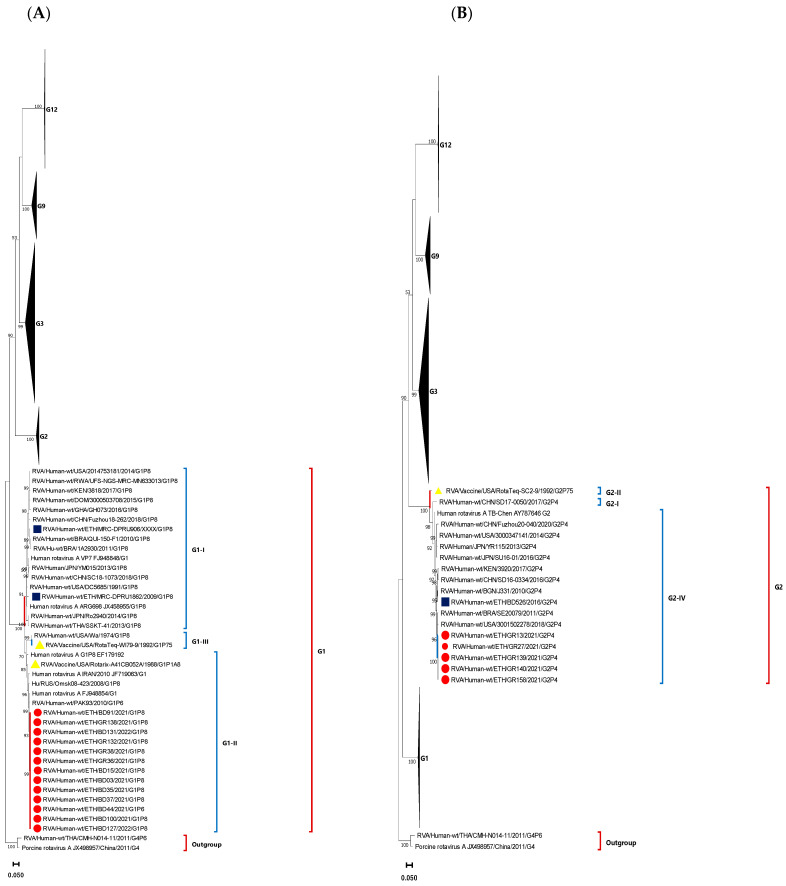
Phylogenetic analysis of the VP7 gene of RVA strains. Maximum-likelihood trees (**A**–**E**) were constructed based on the complete VP7 CDS region gene sequences (981 base pairs). A GTR nucleotide substitution model was used to construct the phylogenetic tree. The human and porcine G4 RVA strains were used as the outgroup. Current Ethiopian strains are marked in red dots, previous Ethiopian Strains are marked in blue squares, and RVA vaccine strains are marked in yellow triangles. Bootstrap values (1000 replicates) of ≥50% are shown at each node. The scale represents the rate of nucleotide substitution per site.

**Figure 7 vaccines-12-00866-f007:**
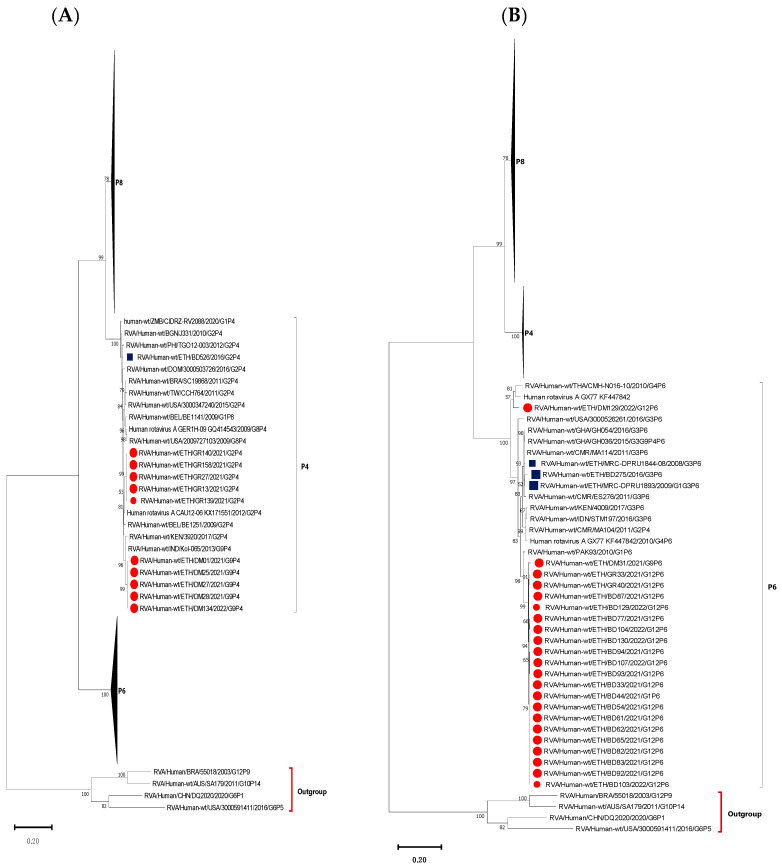
Phylogenetic analysis of the VP4 gene of RVA strains. Maximum-likelihood trees (**A**–**C**) were constructed based on the partial VP4 CDS region gene sequences (810 base pairs). A GTR nucleotide substitution model was used to construct the phylogenetic tree. The human P[1], P[5], P[9] and P[14] RVA strain were used as the outgroup. Current Ethiopian strains are marked in red dots, previous Ethiopian Strains are marked in blue squares, and RVA vaccine P[8] strains are marked in yellow triangles. Bootstrap values (1000 replicates) of ≥50% are shown at each node. The scale represents the rate of nucleotide substitution per site.

**Figure 8 vaccines-12-00866-f008:**
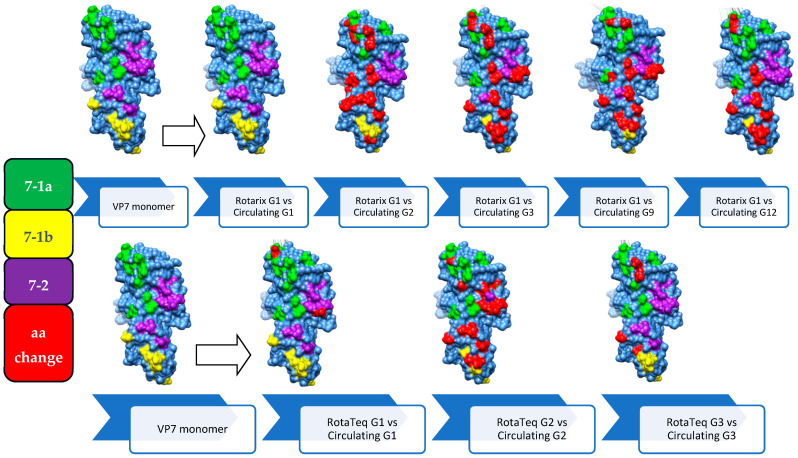
Three-dimensional representation of amino acid substitutions detected in VP7 protein of RVA strains (N = 68). Three-dimensional structure of VP7 monomer (light blue color). Antigenic epitopes are colored in green (7-1a), yellow (7-1b), and purple (7-2). Surface-exposed residues that differ between circulating strains in Ethiopia and strains contained in Rotarix or RotaTeq are shown in red color.

**Figure 9 vaccines-12-00866-f009:**
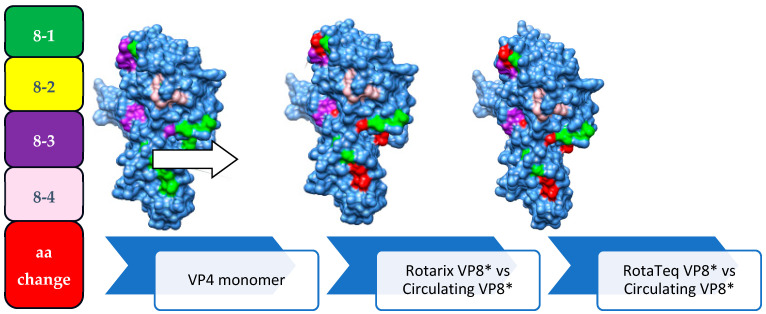
Three-dimensional representation of amino acid changes detected in VP4 (VP8* segment) protein of RVA strains (N = 34). 3D structure of the VP4 monomer (light blue color). Antigenic epitopes are colored in green (8-1), yellow (8-2) not depicted from front, purple (8-3), and light pink (8-4). Surface-exposed residues that differ between circulating RVA strains in Ethiopia and amino acids contained in Rotarix or RotaTeq strains are shown in red color.

**Table 1 vaccines-12-00866-t001:** Sociodemographic and clinical characteristics of the study participants (N = 537).

Variables	Frequency (%)
**Age in months** (mean ± SD) 26.4 ± 15.2 range 2–59 MO	
**Sex**	
Male	308 (57.4)
Female	229 (42.6)
**Residence**	
Urban	494 (92)
Rural	43 (8.0)
**Immunization**	
Yes	524 (97.6)
No	13 (2.4)
**Admission status**	
Yes	41 (7.6)
No	496 (92.4)
**Vomiting**	
Yes	233 (43.4)
No	304 (56.6)
**Child’s thirst status**	
Drink normally	366 (68.2)
Thirsty/drink eagerly	152 (28.3)
Drink poorly/not able to drink	13 (2.4)
N/A	6 (1.1)
**Sunken eyes**	
Yes	83 (15.5)
No	454 (84.5)
**Irritable/restless**	
Yes	133 (24.8)
No	404 (75.2)
**Lethargic**	
Yes	108 (20.1)
No	429 (79.9)
**Mental status of the child**	
Normal	528 (98.3)
Altered	9 (1.7)
**Fever**	
Yes	182 (33.9)
No	355 (66.1)
**Dehydration status**	
No dehydration	496 (92.2)
Some dehydration	40 (7.4)
Severe dehydration	1 (0.2)
**Given IV fluid**	
Yes	57 (10.6)
No	480 (89.4)
**Vesikari clinical severity**	
Mild	198 (36.9)
Moderate	257 (47.9)
Severe	82 (15.2)

IV = intravenous, N/A = not applicable, SD = standard deviation, N = number.

**Table 2 vaccines-12-00866-t002:** Socio-demographic and clinical factors associated with RVA infection among study participants (N = 537).

Variables	RVA		
	Positive (%)	Negative (%)	*p*-Value
**Study locations**			
Gondar	44 (16.9)	217 (83.1)	0.001
Bahir Dar	41 (29.3)	99 (70.7)	
Debre Markos	9 (6.6)	127 (93.4)	
**Age (in months)**			
0–11	17 (19.8)	69 (80.2)	0.04
12–23	46 (23.8)	147 (76.2)	
24–59	31 (12.0)	227 (88.0)	
**Vomiting**			
Yes	54 (23.2)	179 (76.8)	0.002
No	30 (13.2)	263 (86.8)	
**Sunken eyes**			
Yes	22 (26.5)	61 (73.5)	0.019
No	72 (15.9)	382 (84.1)	
**IV fluid given**			
Yes	16 (28.1)	41 (71.9)	0.026
No	78 (16.2)	402 (83.8)	
**Vesikari clinical severity**			
Mild	23 (11.6)	175 (88.4)	0.006
Moderate	49 (19.1)	208 (80.9)	
Severe	22 (26.8)	60 (73.2)	
**Stunting (HFA)**			
Normal	63 (16.8)	313 (83.2)	0.329
Moderately stunted	14 (15.9)	74 (84.1)	
Severely stunted	15 (24.2)	47 (75.8)	
**Weight (WFA)**			
Normal	75 (16.8)	372 (83.2)	0.005
Moderately underweight	10 (15.4)	55 (84.6)	
Severely underweight	7 (50)	7 (50)	
**Wasting (WFH)**			
Normal	75 (16.6)	376 (83.4)	0.007
Moderately wasted	8 (14.8)	46 (85.2)	
Severely wasted	9 (42.9)	12 (57.1)	

IV = intravenous, HFA = height for age, WFA = weight for age, WFH = weight for height.

**Table 3 vaccines-12-00866-t003:** G/P combinations of RVAs in Amhara National Regional State, Ethiopia February 2021–December 2022.

G/P Combinations	Isolates (n = 67)	Proportion (%)
G1P[6]	1	1.5
G1P[8]	13	19.4
G2P[4]	5	7.5
G3P[8]	22	32.8
G9P[4]	5	7.5
G9P[6]	1	1.5
G12P[6]	19	28.4
G12P[8]	1	3
Total	67	100

**Table 4 vaccines-12-00866-t004:** The comparative amino acid sequences analysis of the VP7 antigenic epitopes of the circulating RVA strains (N = 68) with the vaccine strains (Rotarix and RotaTeq).

Strain (No. of Sequences)	Lineage	Epitope 7-1a	Epitope 7-1b	Epitope 7-2
87	91	94	96	97	98	99	100	104	123	125	129	130	291	201	211	212	213	238	242	143	145	146	147	148	190	217	221	264
**Rotarix/A41CB052A/G1P[8]**	**II**	**T**	**T**	**N**	**G**	**E**	**W**	**K**	**D**	**Q**	**S**	**V**	**V**	**D**	**K**	**Q**	**N**	**V**	**D**	**N**	**T**	**K**	**D**	**Q**	**N**	**L**	**S**	**M**	**N**	**G**
**RotaTeq/W179-9/G1P7[5]**	**III**	**T**	**T**	**N**	**G**	**D**	**W**	**K**	**D**	**Q**	**S**	**V**	**V**	**D**	**K**	**Q**	**N**	**V**	**D**	**N**	**T**	**K**	**D**	**Q**	**S**	**L**	**S**	**M**	**N**	**G**
GR36/2021/G1P8 (12)	II	T	T	N	G	E	W	K	D	Q	S	V	V	D	K	Q	N	V	D	N	T	K	D	Q	N	L	S	M	N	G
BD44/2021/G1P6 (1)	II	T	T	N	G	E	W	K	D	Q	S	V	V	D	K	Q	N	V	D	N	T	K	D	Q	N	L	S	M	N	G
**RotaTeq/SC2-9/G2P7[5]**	**II**	**A**	**N**	**S**	**D**	**E**	**W**	**E**	**N**	**Q**	**D**	**T**	**M**	**N**	**K**	**Q**	**D**	**T**	**M**	**N**	**K**	**Q**	**D**	**V**	**S**	**N**	**S**	**R**	**D**	**N**
GR13/2021/G2P4 (5)	IV	T	N	S	N	E	W	E	N	Q	D	T	M	N	K	Q	D	V	D	N	N	R	D	N	T	S	D	I	S	G
**RotaTeq/wi78-8/G3P7[5]**	**II**	**T**	**T**	**N**	**N**	**S**	**W**	**K**	**D**	**Q**	**D**	**A**	**V**	**D**	**K**	**Q**	**D**	**A**	**N**	**K**	**D**	**K**	**D**	**A**	**T**	**L**	**S**	**E**	**A**	**G**
GR03/2021/G3P8 (11)	I	T	N	N	N	S	W	K	D	Q	D	A	V	D	K	Q	D	T	N	N	N	K	D	A	T	L	S	E	A	G
GR10/2021/G3P8 (1)	I	T	N	D	N	S	W	K	D	Q	D	A	V	D	K	Q	D	T	N	N	N	K	D	A	T	L	S	E	A	G
BD34/2021/G3P8 (12)	I	T	T	N	N	S	W	K	D	Q	D	A	V	D	K	Q	D	T	N	N	N	K	D	A	T	L	S	E	D	G
DM01/2021/G9P4 (5)	I	T	T	G	T	E	W	K	D	Q	D	A	I	D	K	Q	N	T	A	D	N	K	D	S	T	L	S	E	S	G
DM31/2021/G9P6 (1)	I	T	T	G	T	E	W	K	D	Q	D	A	I	D	K	Q	N	T	A	D	N	K	D	S	T	L	S	E	S	G
**RotaTeq/Br-B-9/G4P7[5]**	**II**	**S**	**T**	**S**	**T**	**E**	**W**	**K**	**D**	**Q**	**N**	**L**	**I**	**D**	**K**	**Q**	**D**	**T**	**A**	**D**	**T**	**R**	**A**	**S**	**G**	**E**	**S**	**T**	**S**	**G**
**RotaTeq/WI79-4/G6P1A[8]**	**-**	**V**	**N**	**A**	**T**	**E**	**W**	**K**	**D**	**Q**	**D**	**A**	**V**	**E**	**K**	**Q**	**N**	**P**	**D**	**N**	**A**	**K**	**D**	**S**	**T**	**Q**	**S**	**T**	**T**	**G**
BD33/2021/G12P6 (18)		S	T	T	P	D	W	T	N	Q	D	S	V	D	K	Q	D	V	T	N	N	Q	Q	N	S	L	S	E	A	G
BD49/2021/G12P8 (1)		S	T	T	P	D	W	T	N	Q	D	S	V	D	K	Q	D	V	T	N	N	Q	Q	N	S	L	S	E	A	G
BD104/2022/G12P6 (1)		S	T	T	P	D	W	T	S	Q	D	S	V	D	E	Q	D	V	T	N	N	Q	Q	N	S	L	S	E	A	G

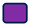
 Different from both Rotarix and RotaTeq. 
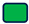
 Different from Rotarix. 
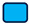
 Different from the closest RotaTeq. 
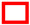
 Sites involved in neutralization escape.

**Table 5 vaccines-12-00866-t005:** Comparative amino acid sequences analysis of the VP4 antigenic epitopes of the circulating RVA strains (N = 65) with the vaccine strains (Rotarix and RotaTeq).

Strain (No. of Sequences)	Lineage	Epitope 8-1	Epitope 8-2	Epitope 8-3	Epitope 8-4
100	146	148	150	188	190	192	193	194	195	196	180	183	113	114	115	116	125	131	132	133	135	86	87	88	89	90
**Rotarix/A41CB52A/G1P[8]**	**I**	**D**	**S**	**Q**	**E**	**S**	**T**	**N**	**L**	**N**	**N**	**I**	**T**	**A**	**N**	**P**	**V**	**D**	**S**	**S**	**N**	**D**	**N**	**S**	**N**	**T**	**N**	**G**
**RotaTeq/WI79-4/G6P1A[8]**	**II**	**D**	**S**	**Q**	**E**	**S**	**T**	**N**	**L**	**N**	**D**	**I**	**T**	**A**	**N**	**P**	**V**	**D**	**N**	**R**	**N**	**D**	**D**	**S**	**N**	**T**	**N**	**G**
GR03/2021/G3P8 (12)	III	D	S	Q	D	S	T	N	L	D	G	I	T	A	N	P	V	D	N	N	N	D	D	S	N	T	N	G
BD34/2021/G3P8 (9)	III	D	S	Q	D	S	T	N	L	N	G	I	T	A	N	P	V	D	N	N	N	D	D	S	N	T	N	G
BD63/2021/G3P8 (1)	III	D	S	Q	D	S	T	N	L	N	G	I	T	A	N	P	V	D	N	R	N	D	D	S	N	T	N	G
GR36/2021/G1P8 (1)	IV	D	S	Q	E	S	T	N	L	T	S	I	T	A	N	P	V	D	S	S	N	D	N	S	N	T	N	G
GR38/2021/G1P8 (10)	IV	D	S	Q	E	S	T	D	L	T	S	I	T	A	D	P	V	D	S	S	N	D	N	S	N	T	N	G
BD49/2021/G12P8 (1)	IV	D	S	Q	E	S	T	D	L	T	S	I	T	A	D	P	V	D	S	S	N	D	N	S	N	T	N	G
**RotaTeq-WI79-9/G1P7[5]**		**G**	**T**	**I**	**G**	**R**	**I**	**T**	**N**	**Y**	**A**	**S**	**E**	**N**	**T**	**S**	**E**	**T**	**S**	**S**	**N**	**A**	**D**	**P**	**T**	**G**	**P**	**G**
BD33/2021/G12P6 (19)	-	D	N	N	E	S	T	N	L	S	E	V	T	A	T	N	Q	S	V	E	N	N	N	P	T	N	Q	Q
DM31/2021/G9P6 (1)	-	D	N	N	E	S	T	N	L	S	E	V	T	A	T	N	Q	S	V	E	N	N	N	P	T	N	Q	Q
BD44/2021/G1P6 (1)	-	D	N	N	E	S	T	N	L	S	E	V	T	A	T	N	Q	S	V	E	N	N	N	P	T	N	Q	Q
GR13/2021/G2P4 (5)	-	D	S	Q	D	S	T	D	L	N	N	I	T	A	S	Q	T	N	N	E	N	S	D	S	N	T	D	G
DM01/2021/G9P4 (5)	-	D	S	Q	D	S	T	D	L	N	N	I	T	A	S	Q	T	N	N	E	N	S	D	S	N	T	D	G

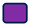
 Different from both Rotarix and RotaTeq. 
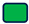
 Different from Rotarix. 
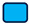
 Different from the closest RotaTeq. 
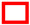
 Sites involved in neutralization escape.

## Data Availability

Data supporting this manuscript are available in the manuscript or online (GenBank).

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
