# Peer review of "Rotavirus A Infection Prevalence and Spatio-Temporal Genotype Shift among Under-Five Children in Amhara National Regional State, Ethiopia: A Multi-Center Cross-Sectional Study"

_vaccines, 2024, doi:10.3390/vaccines12080866_

Round 1

Reviewer 1 Report

Comments and Suggestions for Authors

Rotavirus A (RVA) is still threatening the health conditions of young children worldwide. In the manuscript entitled “Rotavirus A infection prevalence and spatio-temporal genotype shift among under-five children in Amhara National Regional State, Ethiopia: A multi-center cross-sectional study” by Damtie et. al, the prevalence of RVA infection and the genotype distribution among diarrhea children were assessed. They found that despite the high RVA vaccination rate, the prevalence of RVA infection remains significant among diarrheic children in Ethiopia. In addition, an observable shift from G1 to G3, alongside the emergence of unusual G/P genotype combinations were comfirmed. Antigenicity analysis indicated that amino acid substitutions in the circulating RVA strains might attribute to the neutralization escape. These finding emphasizes the importance of continued surveillance and vaccine updating and development in conbating the evolving diversity of RVA strains globally. The authors undertook an interesting work. The ms was well-written and the results were well presented.

There are some weakness that need to be addressed. The manuscript lacks more detailed statistical data of incidence rate at different ages. A map describing the incidence rates and G/P types of RVA in various regions is preferred for readers to understand the prevalence of RVA. Is the prevalence of RVA related to the season? A figure describing the monthly RVA prevalence rate will clearly display it.

Some clerical errors are list below:

1.        According to convention, there needs to be a space between numbers and units, such as 24 hr, instead of 24hr. Please revise the corresponding parts in the main text.

2.        Line 132, “For solid or formed stool 2g of stool was collected,…”, there need to be a comma between stool and 2g.

3.        Line 159, rubes should be tubes?

4.        Line 168, RAN should be RNA?

5.        Line 183, 37.7 μl should be 37.5 μl?

Author Response

Response to Reviewer 1 Comments

1. Summary

Thank you very much for taking the time to review this manuscript. Please find the detailed responses below and the corresponding revisions corrections highlighted in green in the re-submitted MS.

3. Point-by-point response to Comments and Suggestions for Authors

Comments 1: The manuscript lacks more detailed statistical data of incidence rate at different ages.

Response 1: We concur and have included the RVA prevalence data for different age groups on page 9 Table 2.

Comments 2: A map describing the incidence rates and G/P types of RVA in various regions is preferred for readers to understand the prevalence of RVA.

Response 2: We appreciate the reviewer’s comment and have included a map shwoing the prevalence and dominantly circulating RVA genotypes in the three study locations/cities on page 9 Figure 2 in the revised MS.

Comments 3: Is the prevalence of RVA related to the season? A figure describing the monthly RVA prevalence rate will clearly display it.

Response 3: Again, we appreciate this insightful comment and revised the manuscript to address it. Although RVA cases are reported all-year-round, there were some seasonal variabilities in the prevalence of RVA. We have included a new figure 3 depicting RVA distribution by month on page 11 and discussed the results on the 2nd paragraph of the discussion.

Comments 4: According to convention, there needs to be a space between numbers and units, such as 24 hr, instead of 24hr. Please revise the corresponding parts in the main text.

Response 4: Thank you, dear reviewer. We have checked all measurements with units and addressed your concerns.

Comments 5: Line 132, “For solid or formed stool 2 g of stool was collected,…”, there need to be a comma between stool and 2 g.

Response 5: We have modified the statement on page 4, line 133.

Comments 6: Line 159, rubes should be tubes?

Response 6: The spelling/typo error corrected on page 4, line 158.

Comments 7: Line 168, RAN should be RNA?

Response 7: We have corrected the indicated typo error on page 4, line 167.

Comments 8: Line 183, 37.7 μl should be 37.5 μl

Response 7: We have corrected the figure to 37.5 μl on page 5, line 182.

4. Response to Comments on the Quality of English Language

We have additionally reviewed the manuscript and made some typo/spelling corrections throughout and highlighted these changes in green in the revised MS.

Reviewer 2 Report

Comments and Suggestions for Authors

It is  an interesting article. It would be extremely interesting if the authors could comment on what was the incidence of the disease and the circulating  genotypes of the rotavirus before vaccination has started.

Could the authors also comment on the moderate acceptance of the vaccine if it also could affect the circulating genotypes of rotavirus.

What do the author consider to be the best approach for their population in the future and also for which vaccine they would opt

Author Response

Response to Reviewer 2 Comments

1. Summary

Thank you very much for taking the time to review this manuscript. Please find the detailed responses below and the corresponding revisions corrections highlighted in green in the re-submitted MS.

2. Questions for General Evaluation

Reviewer’s Evaluation

Response and Revisions

Does the introduction provide sufficient background and include all relevant references?

Yes/Can be improved/Must be improved/Not applicable

Are all the cited references relevant to the research?

Yes/Can be improved/Must be improved/Not applicable

Is the research design appropriate?

Yes/Can be improved/Must be improved/Not applicable

Are the methods adequately described?

Yes/Can be improved/Must be improved/Not applicable

Are the results clearly presented?

Yes/Can be improved/Must be improved/Not applicable

Are the conclusions supported by the results?

Yes/Can be improved/Must be improved/Not applicable

3. Point-by-point response to Comments and Suggestions for Authors

Comments 1: It would be extremely interesting if the authors could comment on what was the incidence of the disease and the circulating genotypes of the rotavirus before vaccination has started. 

Response 1: Thank you, dear reviewer, for raising the point. RVA prevalence was relatively higher (24%) and dominated by G1P[8] genotype before the introduction of Rotarix vaccine (before 2013). However, the overall prevalence of RVA has dropped a little bit post vaccine introduction and the circulating RVA genotypes tend to shift to G3 and other unusual genotypes such as G12P[6] and G9P[4]. We have indicated it in our discussion on page 27, paragraphs 1 and 4.

Comments 2: Could the authors also comment on the moderate acceptance of the vaccine if it also could affect the circulating genotypes of rotavirus.

Response 2: Though the vaccine might influence the selection of the circulating strains, its contribution in the reduction of the overall prevalence of RVA associated diarrhea remains important. Hence, the currently available vaccine in Ethiopia (Rotarix) has its own role in reducing the overall prevalence of RVA associated diarrhea as well as severity of RVA infection.

Comments 3: What do the author consider to be the best approach for their population in the future and also for which vaccine they would opt.

Response 3: Thank you for pointing out the issue. The current RVA vaccine in Ethiopia (Rotarix) has resulted moderate reduction in the overall prevalence of RVA-diarrhea in Ethiopia. Moreover, the vaccine demonstrated a relatively lower antigenic miss-matches with the circulating strains (in homotypic strains particularly) compared to the RotaTeq vaccine strain. Hence, we would opt for the Rotarix vaccine (currently part of the EPI in Ethiopia). However, as we have clearly indicated in our concluding statement, for optimal vaccine efficacy, we would recommend a continued surveillance, vaccine updating and development, and potentially adapting immunization strategies to better match the evolving diversity of RVA strains in the study area.

4. Response to Comments on the Quality of English Language

Response: No issues were raised on the quality of English language. However, we have reviewed the manuscript and made some typo/spelling corrections and highlighted the changes in green in the latest MS.